# Infection-Related Stillbirths: A Detailed Examination of a Nine-Year Multidisciplinary Study

**DOI:** 10.3390/microorganisms13010071

**Published:** 2025-01-02

**Authors:** Liliana Gabrielli, Matteo Pavoni, Francesca Monari, Federico Baiesi Pillastrini, Maria Paola Bonasoni, Chiara Locatelli, Maria Bisulli, Alessandra Vancini, Ilaria Cataneo, Margherita Ortalli, Giulia Piccirilli, Alessia Cantiani, Simone Ambretti, Fabio Facchinetti, Tiziana Lazzarotto

**Affiliations:** 1Microbiology Unit, IRCCS Azienda Ospedaliero-Universitaria di Bologna, 40138 Bologna, Italy; federico.baiesipillastrini@aosp.bo.it (F.B.P.); margherita.ortalli2@unibo.it (M.O.); giulia.piccirilli@aosp.bo.it (G.P.); simone.ambretti@aosp.bo.it (S.A.); tiziana.lazzarotto@unibo.it (T.L.); 2Microbiology, Department of Medical and Surgical Sciences, University of Bologna, 40138 Bologna, Italy; matteo.pavoni2@unibo.it (M.P.); alessia.cantiani@studio.unibo.it (A.C.); 3Unit of Obstetrics and Gynecology, Mother-Infant Department, University of Modena and Reggio-Emilia, 41124 Modena, Italy; francesca.monari@unimore.it (F.M.); fabio.facchinetti@unimore.it (F.F.); 4Pathology Unit, Azienda USL-IRCCS di Reggio Emilia, 42123 Reggio-Emilia, Italy; mariapaola.bonasoni@ausl.re.it; 5Neonatal Intensive Care Unit, IRCCS Azienda Ospedaliero-Universitaria di Bologna, 40138 Bologna, Italy; chiara.locatelli@aosp.bo.it; 6Obstetric Unit, IRCCS Azienda Ospedaliero-Universitaria di Bologna, 40138 Bologna, Italy; maria.bisulli@aosp.bo.it; 7Neonatal Intensive Care Unit, Maggiore Hospital, 40133 Bologna, Italy; alessandra.vancini@ausl.bologna.it; 8Unit of Obstetrics, Maggiore “C.A. Pizzardi” Hospital, AUSL, 40133 Bologna, Italy; ilaria.cataneo@ausl.bologna.it

**Keywords:** stillbirth, infection, multidisciplinary audit, chorioamnionitis, histology

## Abstract

Background: Although several conditions and specific risk factors have been associated with stillbirth (SB), in most of the cases it is difficult to identify the definitive etiopathology and cause of death. Specifically, the role of infections in SB is still debated. Our aim was to study maternal, placental, and fetal tissues in cases of SB in order to define the causative link between infections and fetal death, through a multidisciplinary clinical audit. Methods: Between 2014 and 2022, microbiological investigations on maternal, placental and fetal samples of SB cases were performed according to a standardized protocol including serology, cultures, and molecular biology. Autopsies and placental examination were mandatory in all SB cases. Results: A total of 182 cases of SB were investigated. Bacteria were detected in 22.2% of vaginal swabs, 65% of placental biopsies, 29% of fetal blood, and 14.1% of oropharyngeal swabs. Vaginal and oropharyngeal swabs were positive for urogenital mycoplasmas in 25.2% and 8.6%, respectively. Positive results of microbiological investigations, in association with histological features suggestive of infection, were observed in six cases, indicating that fetal death was likely related to a bacterial infection. In one case, a high SARS-CoV-2 load was found in the placenta of a SB due to placental abruption. Conclusions: Infections were likely associated with fetal death in 3.8% of cases. Thus, in developed countries, an infection, defined when positive microbiological findings are associated with histological evidence of organ damage, is a minor contributory factor in SB.

## 1. Introduction

Stillbirth (SB) is a sensitive marker of the quality of care during pregnancy and childbirth. Although SB definitions vary worldwide, fetal death is not uncommon, occurring in 3–6/1000 pregnancies in western countries, with a ten-fold increase recorded in low- and middle-income countries [1]. Despite the World Health Organization (WHO) recommendation for a standard definition of SB, heterogeneity in the chronological cut-off point for SB persists among Western European countries and ranges from 22 to 28 weeks of gestation. This heterogeneity limits the ability to conduct cross-national research and compare clinical and epidemiological findings on SB from different countries. Thus, establishing a common cut-off term for gestational age to differentiate SB from spontaneous abortion is the first step in addressing the noted research gaps on SB. We advise adopting the WHO’s definition: ‘‘Birth of a baby showing no signs of life (fetal death) with a gestational age of at least 22 completed weeks. If gestational age is either unknown or not accurate, a birth weight of at least 500 g is required’’. The main justification for adopting the WHO’s definition is the observation that, at current levels of technology, no life is possible prior to 22 weeks of development. The limitation of this definition for universal application is that gestational age is often unknown in developing countries [2].

Establishing the cause of fetal death is paramount in parental counselling, in supporting the mourning process, and in reducing the risk of recurrence with targeted interventions [2,3]. SB prevention must include the collection of appropriate clinical and laboratory data and the implementation of audit systems, which overall lead to the definition of the cause of each case [4].

In Italy, a specific national perinatal audit program is lacking. However, in the northern region of Emilia-Romagna (ER), a Regional SB Audit System, held by a multidisciplinary group, has been instituted since 2014 to analyze data on perinatal mortality [4,5].

SB is associated with several causes of death, each classified differently according to its own setting. Although locations may vary in terms of healthcare system and infection prevalence, fetal death is mostly associated with placenta dysfunction, fetal anomalies, cord accidents, maternal disorders, and infections [6]. Overall, the SB proportion due to infections has been reported as up to 25% [7], varying within different countries. In developed countries, including Italy, the incidence of infections in SB is approximately 10 to 20%, but it is much higher in low-income countries, reaching 30–40% [7,8]. According to the previously mentioned ER Audit System, infections have been recognized as the primary cause of death in less than 10% of the cases [9].

Infections may be associated with a severe systemic disease of the mother (i.e., severe influenza), where fetal death could be due to prolonged fever, respiratory distress syndrome, or systemic reactions, even without the maternal transmission of the microorganisms to the feto-placental unit [7,10,11]. On the other hand, the placenta itself may become severely infected (i.e., malaria), resulting in a reduced fetal blood supply leading to death [12,13]. Finally, SB may be caused by a severe fetal infection directly inducing organ dysfunction/damages as in the case of fetal pneumonia due to *Escherichia coli* (*E. coli*) and/or *Streptococcus agalactiae* (SGB) infection [13].

Therefore, the certain attribution of fetal death to infections can be challenging for several reasons. The positivity of a microbiological specimen in the mother and/or in the fetus is not per se sufficient to define a causal relationship. In fact, despite the evidence of histological chorioamnionitis, in most cases, non-infectious causes may be implicated as alternative factors. In the ER region, a specific microbiological protocol has been established for sample collection and further microbiological investigations. The aim of this study was to highlight the relationships between infections and SB and evaluate the effectiveness of the current protocol in the identification of infectious causes of fetal death.

## 2. Materials and Methods

This was an observational retrospective study on SB cases, which occurred between January 2014 and December 2022 in four birth centers of the ER Region (Italy): IRCCS Sant’Orsola Policlinic of Bologna, Maggiore Hospital of Bologna, Hospital of Bentivoglio, and Hospital of Imola. These four hospitals account for roughly one third of all deliveries in the ER region.

The definition of SB adhered to the guidelines established by the WHO [14], i.e., the demise of a fetus at or beyond 22 weeks (154 days) of gestation, or with a birthweight of at least 500 g in cases where the gestational age was indeterminate. In alignment with WHO guidelines, a late SB was classified as a demised fetus weighing >1000 g and/or at or beyond 28 weeks of gestation, while an early SB was defined as a demised fetus with a gestational age ranging from 22 to 27 weeks.

Demographic and pregnancy data, including delivery, were obtained from the birth certificates database (CedAP), whereas information on the causes of SB was obtained from the Regional SB Audit System database.

The Re.Co.De. classification was applied in order to identify the different causes of death as follows: A fetus, B umbilical cord, C placenta, D amniotic fluid, E uterus, F maternal disorders, G intrapartum event, H trauma, I unclassified [15].

### 2.1. Microbiological Protocol

The microbiological protocol included the following samples:

(a)Maternal vaginal swabs for bacterial and fungal cultures and for urogenital mycoplasmas identification. In our lab, we used a diagnostic assay that was unable to discriminate between *Mycoplasma hominis* and *Ureaplasma urealyticum*/*parvum*.(b)Maternal blood sample for antibody (IgG and IgM) detection for Parvovirus B19, Enterovirus, and Cytomegalovirus (CMV). Serological tests for *Toxoplasma gondii* and *Treponema pallidum* were performed as a routine check during pregnancy; therefore, serological data were collected from pregnancy health records.(c)One placental biopsy (1 × 1 × 1 cm^3^) for bacterial and fungal cultures (in the lab, the sample was cut, and a swab was taken from the inner surface of the tissue).(d)One placental biopsy (1 × 1 × 1 cm^3^) for molecular investigations for Enterovirus, herpes simplex virus (HSV)-1, HSV-2, Parvovirus B19, CMV, and SARS-CoV-2 (after 2020). CMV and Parvovirus B19 polymerase chain reactions (PCR) were performed only in mothers with IgG antibodies for CMV and Parvovirus B19, respectively, using ELITe MGB kits on an ELITe InGenius automated instrument (ELITechGroup, Turin, Italy). The viral load was reported as number of viral copies/µg of DNA/RNA. In case of a positive result on the microbiological test swabs collected from the placenta, the investigations were supplemented by the research of the viral genome on paraffin-embedded tissue. Specifically, 10 μm of paraffin-embedded sections were pre-treated using 160 μL of deparaffination solution, 180 μL of tissue lysis buffer and 20 μL of proteinase K. Viral genome extraction and amplification were performed using ELITe MGB kits on the ELITe InGenius automated instrument (ELITechGroup, Italy).(e)Fetal oropharyngeal swabs for bacterial and fungal cultures and for urogenital mycoplasmas identification.(f)Fetal blood sampling from intracardiac puncture for blood culture.

### 2.2. Fetal Autopsy and Placental Examination

All the SBs underwent a full autopsy. External examination estimated fetal maceration and therefore intrauterine retention, considering the extension of skin slippage in the head, trunk, and limbs. Joint laxity, skull bone overlapping, and sunken eyes were also evaluated. Potentially associated dysmorphic features, anatomical abnormalities, and meconium exposure were also searched for. Internal examination included an extension of postmortem autolysis such as the location of effusions (pleural, abdominal), their type (serous, sero-sanguinous), and their quantification. All the organs were anatomically and pathologically examined and sampled. Sections from the brain, thymus, lungs, heart, liver, pancreas, digestive tract (stomach, small and large intestine), adrenal glands, kidneys, and spleen were all available for histology. In particular, neutrophils within alveoli and along the digestive tract were sought for confirmation of infection including external and internal examination.

The placenta was sampled according to Amsterdam Consensus criteria [16]. Gross evaluation included: placental weight, linear measurements, and thickness; cord length, number of vessels, diameter, coiling index, and insertion into the chorionic plate; membrane integrity, consistency, and color (yellowish for infection, greenish for meconium exposure). In the parenchyma, main lesions such as infarcts, hematomas, thrombo-hematomas, and their extension were recorded. Histologically, membrane meconium exposure, lesions of chorionic vessels and their branches, and abnormalities in parenchymal morphology and maturation were evaluated. Regarding the presence of infection, maternal, and fetal inflammatory response, chorioamnionitis and funisitis, respectively, were staged and graded according to the recently revised Amsterdam Placental Workshop Group Consensus Statement criteria [16,17].

### 2.3. Audit Panel

Every case taken under consideration was discussed in a multidisciplinary Regional clinical audit as established by the Regional protocol. The ER Region’s SB surveillance system is based on the activity of six pre-established area working groups, including the obstetrician-gynaecologist, the neonatologist or paediatrician, the pathologist, the microbiologist, and other specialists as required. Their tasks consisted in periodic audits to analyze the cause of death and the factors associated with it and assess the quality of the care provided, the potential avoidance of the fetal death.

The Birth Commission’s Natimortality Subgroup, composed of the local representatives of each birth center and other experts, collects, for each SB case, the medical records and determines the cause of death. All the cases are recorded, and dubious SBs are discussed during the local audits every four months.

### 2.4. Statistical Analysis

Categorical variables were reported as absolute value and percentage frequency. A univariate analysis was performed to compare values using the Chi-square test or Fisher’s exact test (for variables ≤ 5). A descriptive analysis was also performed for the causes of death. The epidemiological evaluation was carried out both by means of a descriptive analysis and, when possible, by a comparison of categorical variables as indicated above. Statistical significance was acknowledged to a *p*-value < 0.05. Statistical analyses were performed with MedCalc19.1.

## 3. Results

### 3.1. Stillbirth Occurrence

A total of 182 cases of SB were recorded between January 2014–December 2022, therefore accounting for 2.65‰ out of 68,741 live births. The demographic features of the mothers are reported in Table 1.

The primary causes of death according to Re.Co.De. classification are reported in Figure 1. One SB (0.6%) occurred intrapartum. In 33 cases (19.2%), the cause could not be identified despite all the investigations being carried out.

### 3.2. Microbiological Investigations

Maternal blood was analyzed for specific IgG and IgM for Parvovirus B19, Enterovirus, and CMV with an IgG-seroprevalence of 51% (78/153), 9.1% (14/154), and 83.3% (135/162), respectively. Moreover, the available IgG-seroprevalence for *Toxoplasma gondii*, Rubella virus, Measles virus, and *Treponema pallidum* was 23.9% (39/163), 91.7% (121/132), 77.5% (55/71), and 1.8% (3/164), respectively.

Five women were CMV-IgG and CMV-IgM positive. All women had a preexisting immunity, before the pregnancy (three women) or in the first trimester (two women). One woman was IgG and IgM positive for Parvovirus B19, and two women were IgG and IgM positive for *Toxoplasma gondii*. In all these cases, the women were immune before the pregnancy. Two women were positive in the screening test for *Treponema pallidum* but aware of their serological status prior to pregnancy.

### 3.3. Vaginal Swab and Placental Biopsies

Twenty-one of the 126 collected vaginal swabs (16.7%; CI95% 11.2–24.1) were positive for yeasts, with *Candida albicans* as the most prevalent species detected; 28 (22.2%; CI95% 15.9–30.2) were positive for at least one bacterial species, in particular *Enterococcus faecalis* (9; 7.0%; CI95% 3.7–12.8), SGB (10; 7.8%; CI95% 4.3–13.8) and *E. Coli* (6; 4.8%; CI95% 2.2–10.0).

Vaginal swabs were also tested for urogenital mycoplasmas, which were detected in 31 cases (25.2%; CI95% 18.4–33.5).

Regarding placental biopsies, microbiological cultures were performed in 157 cases, and a positive result for yeasts was found in five of them (3.2%; 3 *Candida albicans* and 2 *Candida glabrata*), while bacteria were detected in 103 cases (65.6%; CI95% 57.8–72.6).The most commonly involved species were *Enterococcus faecalis* (36; 22.9%; CI95% 17.1–30.1), *E. Coli* (11; 7.0%; CI95% 4.0–12.1), and SGB (9; 5.7%; CI95% 3.1–10.5). These results are shown in Table 2.

### 3.4. Fetal Samples: Fetal Blood and Oropharyngeal Swab

Fetal blood culture was performed in 140 cases, with two positive results for *Candida albicans* (1.4%; CI95% 0.4–5.1) and 39 for bacteria (27.9%; CI95% 21.1–35.8) (Table 2). In particular, the most identified bacteria were *Enterococcus faecalis* (6, 4.3%; CI95% 2.0–9.0), *E. Coli* (1, 0.7%; CI95% 0.1–3.9), and SGB (6, 4.3%; CI95% 2.0–9.0).

Fetal oropharyngeal swab culture was performed in 149 cases, with a positive result for *Candida albicans* in four (2.7%; CI95% 1.1–6.7) and for bacteria in 22 (14.7%; CI95% 9.9–21.3) with the prevalence of *Enterococcus faecalis* (5, 3.4%; CI95% 1.4–7.6), SGB (6, 4.0%; CI95% 1.9–8.5) and *E. Coli* (7, 4.7%; CI95% 2.3–9.4).

An evaluation of urogenital mycoplasmas was performed in 150 fetal swabs, and 13 of them tested positive (8.7%; CI95% 5.1–14.3) (Table 2).

### 3.5. RT-PCR in Placenta

Only 117 placentas were available for further molecular investigations. All placentas tested negative for Enterovirus and HSV-2, and only one case (0.9%) was positive for HSV-1 with a viral load of 100 copies/µg DNA. This patient was immune for HSV-1 before the pregnancy. HSV-1 DNA was also investigated on paraffin-embedded tissues of fetal lung and brain with a negative result. A histological examination of placenta and fetal organs was not suggestive of infection.

CMV and Parvovirus B19 Real-time PCR was performed only in placentas of women with proven immunity status. In particular, CMV DNA was also investigated in 95/117 (81.2%) placentas, and Parvovirus B19 in 55/117 (47%). CMV and Parvovirus B19 tested negative in all the cases.

The detection of the SARS-CoV-2 genome was carried out on the 26 placentas collected after January 2020. In one case, the mother presented clinical symptoms of SARS-CoV-2 infection (fever, non-productive cough, vomiting, and bilateral low back pain), and an oropharingeal swab tested positive for SARS-CoV-2 just before the delivery. Placental viral load was 4,640,000 copies/µg of RNA, UK variant. At delivery, maternal blood was also SARS-CoV-2 positive with a viral load below 250 copies/mL. Placental abruption was identified as the cause of fetal death, confirmed by post-mortem placental examination as adherent blood clots were observed on the decidua.

SARS-CoV-2 RT PCR performed on fetal paraffin-embedded tissues detected a viral load of 30 copies/20ng of RNA in the brain, but the other organs (stomach, pancreas, liver, thymus, lungs, heart, spleen, intestine, and kidneys) tested negative.

### 3.6. Infectious Causes of SB as Defined by a Multidisciplinary Team

The multidisciplinary review selected six cases in which fetal death was attributed to the identification of a specific microorganism (Table 3).

In particular, in four cases, the infection was supported by acute chorioamnionitis, funisitis, and clusters of neutrophils within the fetal respiratory tract (Figure 2A,B). The other two cases showed severe chorioamnionitis, and one of them also showed associated funisitis, but lungs presented advanced maceration and were not assessable. However, placental and cord findings were sufficient to correlate infection and SB.

### 3.7. Correlation Between Positive Results per Bacteria in Maternal and Fetal Sample and Stillbirth

In Table 4 we compared culture results performed on maternal and fetal samples with histological findings. We observed that among SB cases unrelated to infectious aetiologies, 64.5% of placentas were positive for bacteria. Regarding blood cultures, positive results unrelated to infection-caused SB were 25.7%, and in 25 cases (17.8%), bacteria not responsible for SB had been isolated.

## 4. Discussion

SB still constitutes a worldwide concern, according to the WHO document “Every Newborn: an action plan to end preventable deaths” [18], which prompts the investigation of every SB event in order to reduce unexplained cases and consequently the SB rate.

Accordingly, our protocol included a thorough microbiological investigation on different maternal/fetal samples, and positive findings were found in more than 50% of the specimens. However, the Audit panel attributed the fetal death to bacterial infection in only 3.3% of the cases, and in each one, a positive microbiological result must have been associated with histological inflammatory cells. In the remaining cases, the cause of death was associated with placental complications (including one case SARS-CoV-2 related), umbilical cord accidents, and fetal disorders.

In western countries, the prevalence of infection, as a cause of fetal death, was much higher than our actual data, ranging from 10 to 20% [8].

This discrepancy may be explained by a more effective and accurate multidisciplinary approach able to shed light on the etiopathogenesis of SB events. On the other hand, bacterial positivity was found in samples collected from fetuses whose death was attributed to other factors such as fetal or placental causes. In almost two thirds of the cases with positive placental bacterial results, this finding was most likely the result of contamination during the passage through the birth canal or a healthcare staff handling of the sample. In fact, when comparing SB with and without bacterial aetiology, no statistically significant difference was observed in the case of positive placental culture results. These high results in placental contamination may suggest the necessity of changes in the methodology of sample collection, such as introducing the sub-amniotic swab, as the most sterile environment is the space between the amnion and the chorion [19].

Regarding vaginal swabs, it is reported that bacterial and fungal colonization in asymptomatic women is observed with a variable frequency ranging from 20 to 60% [20,21]. Urogenital mycoplasmas, atypical bacteria, can be commensals of the vaginal flora and not always represent a cause of vaginitis. In women with vaginitis, positivity to urogenital mycoplasmas is reported in approximately 15–40% of cases [22]. This explains the positivity in a quarter of the samples.

In the only infection-linked SB case associated with a positive result for urogenital mycoplasmas, the *E. coli* co-presence did not allow for a correct distinction as to which bacterium was the main contributor to fetal death. In the literature, the correlation between *Mycoplasma hominis* and SB has been reported; however, the one between *Ureaplasma urealyticum*/*parvum* and SB has not [23]. In SB cases not due to infection, a positive result on oropharyngeal swab, though less frequent than placental culture, was observed in 1/10 cases, probably related to contamination. Thus, isolated positivity on the oropharyngeal swab should not be considered as an infection possibly causing SB, but the microbiological results must always be combined and interpreted with the histological findings.

It is known that SGB infection in utero through vertical transmission can result in severe perinatal infection, leading to either SBs or the onset of disease observed at or shortly after birth. In our cases, SGB is the major infectious cause of SB, responsible for half of bacterial-related SBs. Histological investigations confirmed that the cause of death was related to an acute inflammatory reaction secondary to an ascending bacterial infection. Most likely, the microorganism replicated at amnion level, inducing a maternal inflammatory response. Then, the bacterium present in the amniotic fluid was ingested by the fetus together with granulocytes, seen histologically in the lungs. The persistence of the infection induced the fetal inflammatory response initially in the umbilical vein, then in the umbilical arteries (funisitis) and chorionic vessels (Table 3) (Figure 2A,B).

Culture tests on the placentas of non-infection-related SBs gave a high percentage (64%) of positivity, probably due to the contamination of the placenta during the passage through the birth canal. Therefore, a change in the placental sampling is advisable. Prospectively, the subamniotic swab, performed between the amnion and chorion, after their detachment by a sterile blade, could be the best solution, as long as it is carried out in the delivery room.

Regarding viruses, in one placenta, the low HSV-1 load, the absence of an inflammatory infiltrate, and the lack of infection in fetal organs ruled out an infection-related SB, as the viral contamination might have occurred during the passage through the birth canal. Moreover, intrauterine HSV infection is associated with non-specific histological placental findings such as placental infarcts and lymphoplasmacytic villitis [24], none of them detected in the described case.

In the literature, the role of Enterovirus infection as a cause of SB is still debated, although advocated by some authors [25]. Nuovo et al. investigated 21 spontaneous and unexplained abortions and detected coxsackie virus in the placental and fetal tissue of 10 cases. Although the virus was detected in the placenta through molecular biology and immunohistochemistry, the placental parenchyma showed no damage and appeared regular [26]. Batcup et al. reported a case of a maternal infection with coxsackie virus A9 and a SB at 35th gestational week. An investigation of the placenta showed massive perivillous fibrin deposition and focal inflammatory changes [27]. Our in-depth investigations excluded Enterovirus placental infection, suggesting that Enteroviruses are not a major cause of SB, but further studies are needed in order to clarify the potential causal link.

Regarding SARS-CoV-2 infection, its contribution as a cause of SB has been quite hidden during the pandemic phase, but a direct correlation between poor fetal outcome and viral infection emerged in the Alpha and Delta waves [28]. These variants were able to cause placental dysfunctions, increasing the frequency of placental abruption and other adverse consequences [29]. The link between SARS-CoV-2 infection and placental dysfunction was supported by the observation of a reduced risk of SARS-CoV-2 placentitis and SB with maternal vaccination [30]. A recent narrative review confirmed that SARS-CoV-2 can infect placental cells, leading to adverse pregnancy outcomes related to placental dysfunction [31]. It was described that placental trophoblastic cells, in direct contact with maternal blood, show a strong expression of ACE2 throughout pregnancy, supporting the ability of SARS-CoV-2 to infect the placenta via a receptor-mediated mechanism [32]. In our case, the high viral load in the placenta associated with histological damage suggested that there might have been a causal relationship between placental abruption and SARS-CoV-2 infection. The detection of a low viral load in fetal brain is difficult to interpret and could be either related to a contamination during the samples handling or related to a low degree of fetal infection.

Overall, even considering the case with the causal link between SARS-CoV-2 infection and SB, in our clinical series the prevalence of infections-related causes of SB still remained lower than 4% (7/182: 3.8%).

Finally, we found no protozoal or fungal infections related to SB. In the literature, protozoal infections such as malaria, African sleeping sickness, and Chagas disease have been associated with SB in low-income countries. In addition, *Toxoplasma gondii* and fungi rarely cause SB [7,33]. The etiological contribution of protozoal or fungal infections in SB will be evaluated in depth in the future by the evaluation of a higher number of cases.

The main limitation of our study is that all the samples required by the regional protocol were completely available only in a few cases, influencing the accuracy of the microbiological data. In addition, our serological results, due to the low number of SB cases, were useful only to guide the molecular investigation and could not be used for a seroprevalence evaluation. We should also take into account that our results reflect a high-income country background with elevated socioeconomic standards and universal healthcare assistance. Hence, our finding cannot be extrapolated to low- and middle-income countries.

## 5. Conclusions

In our study, we identified infection as a cause of SB only in cases where positive microbiological findings in both the fetus and the mother were associated with histological evidence of placental/fetal damage. This strict criterion ensured the causal link between SB and infection. Our findings indicate that, at least in our casuistry, in high-income countries, infections are a minor contributor to SB, occurring in 3.8% of cases. This underscores the importance of considering a broad range of potential causes when investigating SB and highlights the need for comprehensive multidisciplinary diagnostic approaches to accurately determine the underlying etiology. Furthermore, we suggested a modification of the microbiological protocol applied in order to reduce the high number of positive results for bacteria in placental samples of non-infection-related SB.

## Figures and Tables

**Figure 1 microorganisms-13-00071-f001:**
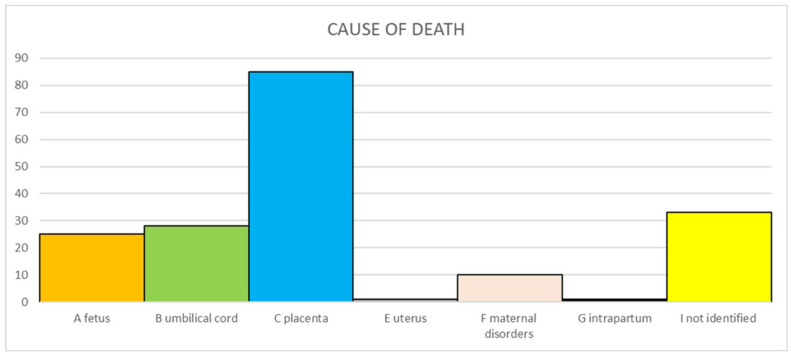
Primary causes of stillbirth according Re.Co.De. classification.

**Figure 2 microorganisms-13-00071-f002:**
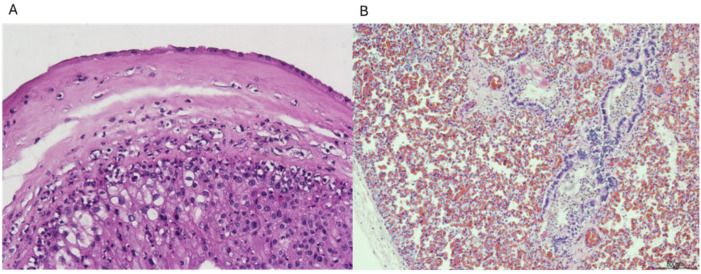
(**A**) Neutrophils in the chorion and sub-amnion (chorioamnionitis); (**B**) Neutrophils in the fetal lung. Hematoxylin and eosin staining, magnification: (**A**) 10 HPF and (**B**) 4 HPF.

**Table 1 microorganisms-13-00071-t001:** Demographic and obstetric features of the 182 mothers who presented with stillbirth.

	Range	N or ±SD	(%; CI95%)
Age (years)	(17–52)	33 ± 6	NA
Italian born		100	(54.6%; 47.2–61.9)
Smokers		24	(13.9%; 9.5–19.9)
Gestational age (weeks)	(22–41)	33 ± 5	NA
Medically assisted reproduction		7	(4.1%; 1.9–8.1)
Primiparity		70	(40.7%; 33.6–48.1)
Previous SB		12	(6.9%; 4.0–11.8)

SB: stillbirth; NA: not applicable; N: number; SD: standard deviation; CI: confidence interval.

**Table 2 microorganisms-13-00071-t002:** Culture results on different types of maternal and fetal samples.

	Bacterial Culture (%Positive)	Yeast Culture (%Positive)	Urogenital Mycoplasmas (%Positive)
Vaginal swab	28/126 (22.2%)	21/126 (16.7%)	31/123 (25.2%)
Placenta	103/157 (65.3%)	5/157 (3.2%)	NA
Fetal blood culture	39/140 (27.9%)	2/140 (1.4%)	NA
Fetal oropharyngeal swab	22/149 (14.7%)	4/149 (2.7%)	13/150 (8.7%)

NA: not applicable.

**Table 3 microorganisms-13-00071-t003:** Infection-related stillbirth.

Case N°	Microrganism	Fetal Histological Examination	Placental and Funicular Histological Examination
104	*Streptococcus agalactiae*	Massive neutrophilic granulocytic infiltrate in the lungs	Chorioamnionitis stage 1/3; grade 1/2 Funisitis stage 1/3; grade 1/2
152	*Streptococcus agalactiae*	Neutrophilic granulocytic infiltrate in the lungs	Chorioamnionitis stage 2/3; grade 2/2 Funisitis stage 2/3; grade 2/2
78	*Enterococcus faecalis*	Neutrophilic granulocytic infiltrate in the lungs	Chorioamnionitis stage 2/3; grade 2/2 Funisitis stage 1/3; grade 1/2
114	Urogenital mycoplasmas/ *Escherichia coli*	Neutrophilic granulocytic infiltrate in the lungs	Chorioamnionitis stage 1/3; grade 1/2 Funisitis stage 1/3; grade 1/2
19	*Enterococcus faecalis*	Fetus with intermediate maceration	Chorioamnionitis stage 3/3; grade 2/2 Funisitis stage 3/3; grade 2/2
97	*Streptococcus agalactiae*	Fetus with severe maceration	Chorioamnionitis stage 3/3; grade 2/2

**Table 4 microorganisms-13-00071-t004:** Correlation between positivity for typical and atypical bacteria in different materials and stillbirth.

	Typical Bacteria Detection (pos/tot)	Positivity/ SB with Bacterial Aetiology	Positivity/ SB with Other Aetiology	*p* Value	Atypical Bacteria Detection (pos/tot)	Positivity/ SB with Atypical Bacterial Aetiology	Positivity/ SB with Other Aetiology
Vaginal swab	28/126 (22.2%)	5/5 (100%)	23/121 * (19.0%)	<0.00001	31/123 (25.2%)	1/1 (100%)	30/122 (24.6%)
Placenta	103/157 (65.6%)	5/5 (100%)	98/152 (64.5%)	0.01	NA	NA	NA
Blood culture	39/140 (27.9%)	4/4 ″ (100%)	35/136 * (25.7%)	0.001	NA	NA	NA
Oropharyngeal swab	22/149 (14.7%)	4/4 ″ (100%)	18/145 * (12.4%)	<0.00001	13/150 (8.7%)	1/1 (100%)	12/149 (8.1%)

SB: stillbirth; pos: positive; tot: total; NA: not applicable; *: statistically significant; ″: 1/5 test not performed.

## Data Availability

The raw data supporting the conclusions of this article will be made available by the authors on request due to the large number of data.

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
