# Peer review of "Infection-Related Stillbirths: A Detailed Examination of a Nine-Year Multidisciplinary Study"

_microorganisms, 2025, doi:10.3390/microorganisms13010071_

Round 1

Reviewer 1 Report

Comments and Suggestions for Authors

Thank you for inviting me to review this manuscript. It is interesting and well-written. I have some comments that could be of use:

·      Line 47: Please define stillbirth (even though it is also mentioned in the methods section). From which week and on is a pregnancy loss considered stiilbirth? Some readers may not be such familiar with this

·      Line 88: Please add the word ‘Italy’, after Inola. It might seem obvious by the affiliation of the authors, but not be for all

·      Lines 121-124: No need to change lines/paragraphs here

·      An important comment has to do with ethics approval. I couldn’t find a statement regarding the approval of the study from a hospital, the ministry, or another regulatory body. I think this is of critical importance

·      Table 1 and 4: Each table and figure should have a footnote explaining all abbreviations in full

·      Tables 1, 2 and 4: Please replace // with ‘NA’ and define that as ‘not applicable’ at the footnote

·      Subsection 3.2: It is not clear if the antibodies detected were IgG (probably irrelevant, indicating previous exposure) or IgM (probably relevant, indicating current exposure). This should be addressed, since IgG would be bystander and IgM could imply causality with the stillbirth

·      It would be better to say E. coli, rather than EC. This applies to all microorganisms mentioned in this manuscript in my opinion

·      Figure 2 and the corresponding text should be moved to the results section

·      A paragraph mentioning the limitations of this study should be added at the end of the discussion section, just before the conclusions. For example, this study represents only developed countries, and there is no mention of IgM vs IgG for some infectious causes (see comment above)

Author Response

Dear colleague,

we would like to express our gratitude for your review of our manuscript. Your constructive feedback and suggestions have contributed to improving the quality and clarity of our work. We appreciate the time and effort you have dedicated to this review process.

Find our point-by-point answers:

Thank you for inviting me to review this manuscript. It is interesting and well-written. I have some comments that could be of use:

  • Line 47: Please define stillbirth (even though it is also mentioned in the methods section). From which week and on is a pregnancy loss considered stiilbirth? Some readers may not be such familiar with this

Thanks for the suggestion, we added this paragraph in the text: Despite the existence of a World Health Organization (WHO) recommendation for a standard definition of SB, heterogeneity in the chronological cut-off point for what is con-sidered a SB persists among Western European countries, and ranges from 22 to 28 weeks of gestation. This heterogeneity limits the ability to conduct cross-national research and compare clinical and epidemiological findings on SB from different countries. Thus, estab-lishing a common cut-off value for gestational age to differentiate SB from spontaneous abortion is the first step to addressing the noted research gaps on SB. We advise adopting the WHO’s definition: ‘‘Birth of a baby showing no signs of life (fetal death) with a gesta-tional age of at least 22 completed weeks. If gestational age is either unknown or not accu-rate, a birth weight of at least 500 grams is required.’’ The main justification for adopting the WHO’s definition is the observation that at current levels of technology, no life is pos-sible prior to 22 weeks of development. The limitation of this definition for universal ap-plication is that gestational age is often unknown in developing countries.

  • Line 88: Please add the word ‘Italy’, after Inola. It might seem obvious by the affiliation of the authors, but not be for all

Thanks for the suggestion, done.

  • Lines 121-124: No need to change lines/paragraphs here

We changed line for each sample needed, one for fetal oropharyngeal swabs and one for blood samples. We also created a pointed list a to f.

  • An important comment has to do with ethics approval. I couldn’t find a statement regarding the approval of the study from a hospital, the ministry, or another regulatory body. I think this is of critical importance

We have added at the end of the manuscript our declaration for Ethical approval.  Hereafter, you can find our statements regarding the non-necessary obtainment of approval from our Ethical Committee:

According to Italian legislation ('art. 2, comma 7, legge n. 3 del11/01/2018), “retrospective data collections, as part of improvement of healthcare quality and included in non-profit institutional projects like audit activities, do not require ethical committee approval".

Regarding the Informed consent we declare as follows:

Informed consent for diagnostic work-up was not required because in Italy diagnostic investigation is mandatory by law in case of SB (D.M. 7/2014 and D.P.C. 170/99). Patient and fetus privacy was ensured during the phase of data collection and analysis.

  • Table 1 and 4: Each table and figure should have a footnote explaining all abbreviations in full

Thanks for the suggestion, done.

  • Tables 1, 2 and 4: Please replace // with ‘NA’ and define that as ‘not applicable’ at the footnote

Thanks for the suggestion, done.

  • Subsection 3.2: It is not clear if the antibodies detected were IgG (probably irrelevant, indicating previous exposure) or IgM (probably relevant, indicating current exposure). This should be addressed, since IgG would be bystander and IgM could imply causality with the stillbirth

Thanks for the suggestion, we have improved this section.

  • It would be better to say E. coli, rather than EC. This applies to all microorganisms mentioned in this manuscript in my opinion

Thanks for the suggestion, done.

  • Figure 2 and the corresponding text should be moved to the results section

Thanks for the suggestion, we moved this data to the result section and we discussed it in the discussion.

  • A paragraph mentioning the limitations of this study should be added at the end of the discussion section, just before the conclusions. For example, this study represents only developed countries, and there is no mention of IgM vs IgG for some infectious causes (see comment above)

Thanks for the suggestion, we have added a limitation of the study section at the end of the discussion section.

Reviewer 2 Report

Comments and Suggestions for Authors

Stillbirth, defined as the death of a fetus at or after 22 weeks of gestation or with a birthweight of at least 500 grams, remains a significant public health concern worldwide. Despite advancements in prenatal care, the causes contributing to these tragic outcomes are often overlooked, including both infectious and non-infectious factors. Infections account for a significant amount of such cases, ranging from 10% in developed countries, to 40% in low-income ones. The aim of this study was to highlight the importance of infections in the circumstances of stillbirths by using a specific microbiological protocol, which included maternal vaginal swabs and blood samples, placental biopsies and microbiological testing, as well as fetal oropharyngeal swabs and blood sampling. Even in cases of stillbirth with other etiologies than infectious diseases, over half of the placentas that were examined had positive culture results for bacteria, as well as a quarter of the blood cultures.

The study provides valuable information, but there are a few changes that need to be made, which I’ve listed based on the line #:

29 - remove the last “and” and form a new sentence from “Specifically” onwards

38 – rephrase “cultural positive results” →  “positive results of microbiological investigations”

43 – rephrase “causal” → “is a contributing cause”

55 – rephrase “acausal definition of each case” – unclear

55 – reference 4 is unclear; the provided link doesn’t redirect to what seems to be cited in the article

58 – could not find the relevant information cited in the article for reference 5

59 – replace “recognizes” →  “associates”

60 – please elaborate on the relevance of location in this specific phrase; 

62 – rephrase “distributed within” →  with rates varying between 10-25% within different countries in one study; the next sentence gives different data that goes above the previously mentioned 25% 

65 – rephrase “above reported” → “previously mentioned”

64 – after analyzing the article in reference 8, it doesn’t include the specific data you mention, but rather further redirects to other studies; please add those as references if you see fit

66 – in the cited article, the leading cause of death in intrapartum SB was infection (41.1%); later in the same cited article, infections are considered another significant cause of death, with almost 12%; fetal infection, according to Table 2 of the cited article, accounts for 6.1% of the cases – maybe specify this instead

68 – rephrase → “due to maternal symptomatology, such as prolonged fever, …” 

67-70 – could not find the information in the cited articles

71-74 – could not find the information in the cited article #13

77-79 – rephrase → “in spite of histologically proven chorioamniotitis being a primary cause of stillbirth in most cases, non-infectious causes may be implicated as alternative factors”

80 – replace “set” → “established”

81-82 – rephrase → “the aim of this study was to highlight the relationships between infections and SB. To do so, the effectiveness of this protocol in the identification of infectious causes of fetal death was evaluated.”

85 – add: “sb cases which occurred”

92-94 the stillbirths were defined as the demises of fetuses, not as the fetuses themselves

96 Punctuation error - comma after (CedAP), 

98- add: “in order to identify”

107-108 rephrase → “were also tested in patients who had not gone through routine screening during pregnancy”

110- replace: “from the inner surface of the tissue”

a list of Abbreviations in the introduction could be useful, such as PCR 

116 – rephrase → “positive result on the microbiological test swabs collected from the placenta”

130 – replace/rephrase → “to the recently revised Amsterdam Placental Workshop Group Consensus Statement criteria”

143 – replace “occurred” →  “were recorded between January 2014-dec 2022, therefore accounting for 2.65%o out of 68741 live births”

149 – replace →  “despite all the investigations being carried out”

163 – rephrase → “urogenital Mycoplasmas, which were detected in 31 cases”

167 – replace: “grown” →  was detected/grew

171 – page formatting so that Table 2 title is on the same page as the actual table; same for table 3 on line 208

185 – replace → “were negative/positive” -> “tested negative/positive”

187-188 – add+rephrase “was also investigated … yielding a negative result”

190-191 – rephrase → “placentas of women with proven immunity status”

197 – rephrase “a positive swab just before delivery tested positive”

197 – replace → “placental viral load turned out of” -> “...was”

211 – rephrase → “in one funisitis” – “one of them also associated funisitis”

Reference 18 – please follow citation guidelines for line 371

228 – take out the word “often”

230 – replace “only in” →  “in only”; “a positive microbiological result was necessary to be associated with histologically proven inflammatory cells”

238 – rephrase “fetal samples … “ →  “samples collected from fetuses whose deaths were attributed …”

239 – rephrase → “two thirds of the cases with positive placental bacterial results”

242 – rephrase →  “may suggest the necessity of changes in the methodology of sample collection, such as introducing”

243 - Punctuation error – comma after swab

245 – replace “considering” -> “regarding”

246 – the cited article (20) reports findings on Candida, which leads to fungal colonization; replace “ranging” -> “ranging from”; couldn’t find the cited information in reference #21

246 - remove “especially”

247 – add: not always “represent a” cause of vaginitis

249 – reference #22 notes an 11.5% positivity rate for mycoplasma hominis for females; mollicutes co-infection (with ureaplasma) has a positivity rate of 19.1%

253 - please rephrase -“ the correlation between MH and SB has been reported in literature, however the one between UUP and SB has not”; there has been a prospective, observational study on 2437 patients that described UUP as being the most detective organism when it came to SB - 6.4% of SBs https://doi.org/10.1111/1471-0528.17479

262 - punctuation error: comma after SB, and also on line 265 after level

271 - page formatting error, please make it so that figure 2 text is under the actual figures

282 - the main author’s name is Batcup

285 - remove “whilst”

303 - replace “level” → “degree”

307-308 - for the protozoal infections, not all of them are discussed in the cited article, or they are considered not well enough understood due to low data; couldn’t find data about fungal infections in the cited article

312- rephrase → “we considered infections to be a causal factor for SB”

Comments on the Quality of English Language

The English language should be improved.

Author Response

Dear colleague,

we would like to express our gratitude for your review of our manuscript. Your constructive feedback and suggestions have contributed to improving the quality and clarity of our work. We appreciate the time and effort you have dedicated to this review process.

We have corrected the manuscript following your suggestions and we have modified references in order to better clarify data.

Reviewer 3 Report

Comments and Suggestions for Authors

Data that associate seasonal microbiology with the risk of stillbirth (SB) in an Italian region. Some comments could improve the results.

- Introduction. Concisely, the objective of the work is clear. 

- Material and methods. Report more data in section 2.2 and 2.3. Also, report to the ethics committee. What do you mean by regional protocol? what was wanted?

- In the statistical section, it is necessary to report the statistical software and the statistical potential used.  

- Results. Figure 1 can be reported in the text. Section 3.7 shows no correlations. Contrast p-values ​​should be reported in the tables.

- Discussion. According to the reported p-values ​​in the associations, the effect size of SB and infections (at least as a whole) could be calculated to determine what proportion of SBs are due to an infection.

Author Response

Dear colleague,

we would like to express our gratitude for your review of our manuscript. Your constructive feedback and suggestions have contributed to improving the quality and clarity of our work. We appreciate the time and effort you have dedicated to this review process.

Find our point-by-point answers:

Data that associate seasonal microbiology with the risk of stillbirth (SB) in an Italian region. Some comments could improve the results.

- Introduction. Concisely, the objective of the work is clear. 

- Material and methods. Report more data in section 2.2 and 2.3. Also, report to the ethics committee. What do you mean by regional protocol? what was wanted?

Thanks for the suggestion, we have improved these sections.

- In the statistical section, it is necessary to report the statistical software and the statistical potential used.  

Thanks for the suggestion, we have improved this section.

- Results. Figure 1 can be reported in the text. Section 3.7 shows no correlations. Contrast p-values ​​should be reported in the tables. Discussion. According to the reported p-values ​​in the associations, the effect size of SB and infections (at least as a whole) could be calculated to determine what proportion of SBs are due to an infection.

Thanks for the suggestion, we have indicated p values in the table. We have also improved our discussion regarding this findings.

Round 2

Reviewer 1 Report

Comments and Suggestions for Authors

The manuscript has been improved

Author Response

Dear Colleague,

we would like express our gratitude for your support.

Best regards

Reviewer 2 Report

Comments and Suggestions for Authors

121 defined as a "demised" fetus

137 remove "Whereas"

175 "all the organs were sampled and examined histologically" is repeated, similar sentence on lines 170-171

198 whose task"s were" to ...

202 "every four months" towards either the beginning or the end of sentence

218 "‰" instead of %o

237 explain what the "(2)" stands for

251 "regarding" instead of "considering"

287 "an oropharyngeal"

294 "tested" instead of "were all"

298 "infection" instead of "infectious"

312-313  "positive results unrelated to infection-caused SB"

342 "SB" instead of sb

344 "positive placental culture results" instead of placenta positivity

368 responsible "for"

370 spelling error- "related"

381 "infection-related SB" instead of SB related infection

389 describe the histological non-specific changes found in Nuovo's study

424 replace “directly translated in” with “extrapolated to”

434 - "still represents a minor causal factor of SB"

Table captions should appear above the table. Figure captions should appear below the figure

Reiterate in the final part the necessity of better protocol for sample collection, in order to avoid contamination. Moreover, you could formulate a more specific conclusion: firstly, replace “defined causative infection of SB (...)” with “considered infection to be the cause of SB (...)” and then add more of your findings, such as the prevalence of specific bacteria in infection-related SBs and the lack of protozoal/fungal causes 

Punctuation missing throughout the paper - such as 276 comma after “HSV-1”; 393 comma after “infection”; 398 comma after "dysfunctions"

Try to rephrase the conclusion section. Although it must stand out clear and simple, it is too short and unexpressive.

Author Response

Dear colleague,

we would like to thank you for the support in the revision of our paper.

We have corrected the manuscript following your suggestions.

We have deepened the description of the results reported in the paper by Nuovo adding this sentence: "Although the virus was detected in the placenta through molecular biology (viral RNA after in situ amplification of the cDNA) and immunohistochemistry, the placental parenchyma showed no damage and appeared regular."

We have also improved the final part of the discussion and finally we have rephrased the conclusion section.

Best regards